# Fat Deposition and Fat Effects on Meat Quality—A Review

**DOI:** 10.3390/ani12121550

**Published:** 2022-06-15

**Authors:** Madison Schumacher, Hannah DelCurto-Wyffels, Jennifer Thomson, Jane Boles

**Affiliations:** Animal and Range Sciences Department, Montana State University, Bozeman, MT 59717-2900, USA; madison.schumacher15@gmail.com (M.S.); hannah.delcurto@montana.edu (H.D.-W.); jennifer.thomson@montana.edu (J.T.)

**Keywords:** fat deposition, carcass yield, breed, fat transcripts, inflammation

## Abstract

**Simple Summary:**

Animal fat deposition has a major impact on the meat yield from individual carcasses as well the perceived eating quality for consumers. Understanding the impact of livestock production practices on fat deposition and the molecular mechanisms activated will lead to a better understanding of finishing livestock. This enhanced understanding will also lead to the increased efficiency and improved sustainability of practices for livestock production. The impact of fat storage on physiological functions and health are also important. This review brings together both the production practices and the current understanding of molecular processes associated with fat deposition.

**Abstract:**

Growth is frequently described as weight gain over time. Researchers have used this information in equations to predict carcass composition and estimate fat deposition. Diet, species, breed, and gender all influence fat deposition. Alterations in diets result in changes in fat deposition as well as the fatty acid profile of meat. Additionally, the amount and composition of the fat can affect lipid stability and flavor development upon cooking. Fat functions not only as a storage of energy and contributor of flavor compounds, but also participates in signaling that affects many aspects of the physiological functions of the animal. Transcription factors that are upregulated in response to excess energy to be stored are an important avenue of research to improve the understanding of fat deposition and thus, the efficiency of production. Additionally, further study of the inflammation associated with increased fat depots may lead to a better understanding of finishing animals, production efficiency, and overall health.

## 1. Introduction

Fat deposition is an important aspect of meat quality. Meat quality can be defined in numerous ways, ranging from product yield to eating satisfaction. These are examples of different qualities identified in unique portions of the production chain. Producers might view meat quality/fat deposition as the appropriate time to harvest an animal or the condition an animal might be in. In contrast, the processor may view fat deposition as either a problem, as it must be removed from a carcass, or essential, as the ingredient in a processed product. Consumers will have an even more different opinion, with some preferring leaner meat, while others may seek out meat cuts that have more fat. These additional requirements of the various industry segments make it important to understand why the targets exist and how fat is deposited throughout an animal’s life. Fat accumulates as the animal matures and is deposited in various fat depots. Berg and Butterfield [1] reported that fat accumulation occurred after the relative growth of muscle decreased and continued to increase while bone growth decreased. The growth rate of fatty tissues varies widely depending on the location and growth stage [2].

An animal’s growth is often described as a sigmoidal curve indicating the change of weight over time [3,4,5]. The body’s composition also changes with fat deposition occurring later after muscle and bone growth have slowed [1,3,4]. Robelin [2] stated that the lipid content of body tissues increases from 25% around birth to 50–75% in adults, while the water and proteins decrease as a percentage of the whole body. Furthermore, the deposition of fat occurs when the energy consumed is greater than the requirements of the animal [6]. When the energy required for bone and muscle growth is reduced as the animal reaches mature size, the energy will be stored as fat. Several factors such as species [7], genetics, or breed [8,9,10,11,12,13,14], sex [9,10,14,15,16], and environmental factors [15] can influence fat development at various points of an animal’s life.

Fat deposition happens in specific depots that are common among all mammals. These depots are found in the abdominal cavity, intermuscularly (between muscles), subcutaneously, and intramuscularly (within muscles). The internal fat, especially around the internal organs, is the first to deposit, followed by intermuscular fat, subcutaneous fat, and finally, intramuscular fat [2,17,18,19]. There is a difference in the proportion of each fat depot depending on the species and age of the animal, as well as the energy intake. For example, pigs have more subcutaneous fat at about 70% of their total body fat and less abdominal fat than both sheep and cattle. In mature grass-finished beef steers, Pethick and Dunshea [20] reported that subcutaneous fat makes up about 15% of their total body fat, intermuscular about 23%, and intramuscular about 14%.

Kempster [7] reviewed the literature on fat distribution in cattle, sheep, and pigs. Cattle had faster subcutaneous fat deposition than intermuscular fat, and internal fat had an intermediate rate of fat deposition. Swine, however, deposited internal fat more quickly, followed by subcutaneous fat, with the intermuscular fat deposition rate being the slowest. Lastly, sheep had a similar rate of subcutaneous fat and internal fat deposition, with intermuscular fat deposited at a slower rate. Differences seen between species may reflect the differences in diets and digestive tracts. The fatty acid composition of the diet is reflected in the composition and distribution of fat in non-ruminant animals. Along with this, genetic differences can be found within species. Iberian pigs are well known for their high fat deposition. Pena and colleagues [21] identified six genes associated to variations in the fatty acid composition. In contrast, ruminant animals consume more forage-based diets with a lower total fat content and the composition of stored lipid reflects a microbial hydrogenation rather than dietary fat composition [22].

Fat is made up of triglycerides. The triglyceride has a glycerol backbone, with three fatty acids making up the rest. The triglyceride’s fatty acids vary by carbon chain length and the number of saturated or unsaturated bonds within the carbon chain [23]. Fatty acids in meat are predominantly palmitic, stearic, oleic, palmitoleic, linoleic, linolenic, and arachidonic [23,24]. The fatty acid composition in adipose tissue affects the firmness of the fat. Longer chain fatty acids result in higher melting points, while fatty acids with more unsaturated bonds have lower melting points. Composite fatty acids melt between 25 °C and 50 °C, saturated fats melt at higher temperatures, and polyunsaturated fats (PUFAs) melt at lower temperatures. This translates to different firmnesses of the fats between species.

The differences in fat accumulation and composition between species are partly due to the differences in digestion processes. Fatty acids in non-ruminant fat and muscle reflect the fatty acid composition of their diets [23]. Ruminant fatty acid composition is influenced by biohydrogenation in the rumen [23]. In ruminants, lipids entering the rumen must go through lipolysis, where the lipases hydrolyze the ester bonds in complex lipids and result in the release of fatty acids. The unsaturated fatty acids are converted to saturated fatty acids by an isomerization from *cis* to *trans* fatty acid intermediates, followed by hydrogenation of the double bonds (Figure 1) [25]. The rate at which lipolysis and biohydrogenation occur is dependent on the type and amount of fat delivered to the rumen, as well as the ruminal pH [26,27,28].

Feed intake, as well as the chemical composition of feed, can affect fat deposition in livestock [29]. Feeding non-ruminant livestock dietary oils changes the fatty acid composition of the subcutaneous fat, altering the adipose tissue’s melting point and overall firmness [24], resulting in softer, more unsaturated carcass fat. The supplementation of unsaturated fatty acids to ruminants is a little more difficult due to the biohydrogenation of the rumen converting the unsaturated fats to more saturated fat [30], ultimately resulting in harder carcass fat. Fatty acids in ruminants are degraded to monounsaturated and saturated fatty acids, leaving roughly 10% of the dietary fatty acids available for incorporation into adipose tissue [24]. However, if a diet extremely rich in unsaturated fatty acids is fed to ruminants, there is a slight chance that the 10% of fatty acids available to lipid tissue could result in softer fat [31]. This change is concomitant with changes in the rumen microbiota. Specifically, the increased oil content decreases ruminal cellulose degradation and volatile fatty acid concentration. This is mediated by an increase in small cocci and a decrease in small rods within the rumen microbiota [22].

Palm kernel oil (more saturated fatty acids) and palm oil, used as a replacement for soybean oil in swine diets, showed altered fatty acid composition. The palm kernel supplemented animals had higher quality sliced bacon but harder subcutaneous fat than animals consuming palm oil and soybean oil, reflecting the higher saturated fatty acid content in the supplemented oil. Animals fed palm oil had softer fat than the palm kernel oil-fed animals, but still were not as soft as the fat from animals fed soybean oil. Both oils, palm kernel oil and palm oil, increased the amount of saturated fatty acids in the muscle, reducing the nutritional ratio of polyunsaturated to saturated fatty acids to below 0.4 [32,33]. Similarly, Smink et al. [34] found that the palmitic acid concentration in breast muscle increased in broilers-fed diets containing palm oil compared to those provided with sunflower oil.

Feeding oils can result in some changes in beef; however, they will not yield the same results as in pork or poultry. While some differences may be observed in the fat composition of beef cattle, they will not be as extreme due to the biohydrogenation of the rumen. Scollan et al. [35] evaluated the effect of feeding beef cattle grass silage with different sources of lipids: Megalac (16:0), lightly bruised whole linseed (18:3*n*-3), and fish oil (20:5*n*-3 and 22:6*n*-3). They concluded that the type of lipid fed to the animals did not influence feed intake, growth rate, cold carcass weights, or carcass fatness. However, omega-3 concentrations within the muscle and adipose tissue significantly increased in cattle fed bruised linseed and fish oil [35,36]. These findings were similar to those of Manner, Maxwell, and Williams [37], where values for 18:2*n*-6 and 18:3*n*-3 from the control group were between the values found for grass- and grain-finished cattle. Additionally, muscle from grass-finished animals had higher levels of mono- and poly-unsaturated fatty acids than muscle from concentrate-based systems [38]. Meat from grass-fed steers will have increased levels of linoleic acid, trans vaccenic acid, and omega-3 fatty acids [39]. Furthermore, meat from pasture-fed bulls had higher poly-unsaturated fatty acids when compare to bulls fed concentrate [40].

Increasing the concentration of unsaturated fatty acids in muscle makes it more susceptible to lipid oxidation. The initiation of the oxidation of the lipids starts with the extraction of allylic hydrogens or ones adjacent to a double bond [23]. This is followed by forming a reactive oxygen species or an •OH radical or singlet oxygen. The initiation of lipid oxidation results in lipid radicals, propagating the reaction. This continues until two lipid radicals react to form a non-radical compound [23]. This process results in various aldehydes, ketones, alcohols, esters, and carboxylic acid. These compounds influence the flavor of the meat. Thermal oxidation results in desirable volatile profiles, while the auto-oxidation of raw and cooked meat products leads to off-flavor development. The thermal oxidation process, along with Maillard reactions, helps develop the characteristics, aromas, and flavors of cooked meat. The fatty acid profile of the meat gives the meat the characteristic cooked flavor [23].

Species flavor was initially attributed to the lipid degradation during the cooking of species-specific fatty acids [41]. However, phospholipids are a significant portion of meat and will also affect the formation of flavor compounds [41]. Volatile compounds resulting from lipid degradation will impact the flavor of the meat. These compounds can be simple aldehydes, alcohols, and ketones, but some aromatic (cyclical structure) compounds have also been reported, such as lactones and alkylfurans [41]. The volatile compounds generated from lipids during cooking are the major contributors to the flavor of cooked meat [41].

## 2. Genetics or Breed Effect on Fat

Many researchers have observed fat deposition differences in breeds that selection can also affect. Berg and co-workers [10] compared the carcass composition of seven different beef breeds reared to a specific age, and the breeds differed in the muscle, fat, and bone composition of the carcasses. Furthermore, at a standard carcass weight, larger framed breeds, such as Chiannia and Blonde d’Aquataine, resulted in carcasses with less fat than those from Danish Red and Hereford. These researchers attempted to develop a biologically sound statistical methodology for group comparisons of growth patterns and carcass composition. They suggested that when serial slaughter was incorporated into the statistical methodology, there was an opportunity to examine the growth patterns of individual tissues by using regression and covariance analysis [10]. Tess et al. [13] found significant differences in carcass fat when pig lines were selected for fat accumulation. Quiniou et al. [14] also reported that the pigs’ breed composition affected the subcutaneous fat measurements.

Similar results have been reported for sheep. Fourie et al. [9] and Gothoh et al. [19] reported that genetic predisposition or breed and mature size influence when and how much fat is accumulated at a given age in lambs. Fourie et al. [9] reported differences in the fat content in carcasses from two different breeds of lambs representing different frame types. Carcasses from Romney lambs had more fat than carcasses from Southdown and Southdown cross lambs. Taylor and co-workers [42] evaluated both males and females from Soay, Welsh Mountain, Southdown, Finnish Landrace, Jacob, Wiltshire Horn, and Oxford Down sheep breeds. Differences in the fat deposition were observed with carcasses from females depositing more fat in all areas compared to intact males. Selection can also impact the fat deposition in lambs. The fat measurements in the domestic breeds evaluated varied, but had about twice as much fat as the feral or less selected breeds. Butterfield and co-workers [11] evaluated large and small strains of Merino rams. They concluded that the proportion of muscle and bone was similar, but there was a slightly greater proportion of fat in the larger strain of sheep.

### 2.1. Gender

In general, for beef animals compared at similar carcass weights, heifers will be fatter than castrate males, which are both fatter than intact males [43]. However, Berg et al. [43] also stated that comparisons between sexes are probably minor if the comparisons are made at equal fatness. Furthermore, Maniaci et al. [44] reported muscles from mature cows had more fat compared to muscles from bulls that were reared in confinement or grazed. Data for the effect of gender on carcass fat content in sheep are similar to beef. Taylor et al. [42] observed increased fat in ewes compared to rams, while Butler-Hogg and co-workers [12] also observed that ewe carcasses required more trimming of subcutaneous fat than ram carcasses. In contrast, swine have fewer differences in fat deposition between sexes. Davies et al. [15] reported that sex differences (castrate male compared to gilt and intact male) had less effect on pork carcasses than what is seen in other species. Woodworth and colleagues [45] conducted a meta-analysis of 34 different peer-reviewed papers published since 2000, including 16,000 animals. The results suggest that gilts have 11.7% less backfat and 4.5% increased lean percentage compared to barrows.

### 2.2. Environment

Environmental factors that influence metabolism can affect fat deposition. Chronic heat stress reduces beta oxidation and positively influences lipid deposition as a method of reducing thermogenesis [46]. Furthermore, Heng and colleagues [47] reported maternal exposure to heat stress altered the expression of genes associated with lipid metabolism and storage, resulting in the increased fatness of piglets produced from heat-stressed dams. Kouba et al. [48] concluded that growing pigs chronically exposed to heat stress had an enhanced lipid metabolism in both the liver (VLDL production) and the adipose tissue (lipoprotein lipase activity). As a result, plasma triglyceride uptake and storage are facilitated in the adipose tissue, which results in greater fatness. In dairy cattle, Hao [49] reported that heat-stressed animals had an increased lipogenic capacity but reduced lipolysis. Furthermore, heat-stressed animals had reduced levels of non-esterified fatty acids. Lu et al. [50] evaluated the effect of heat stress on fat deposition in two genetic types of chickens. Their results indicated that the impact of heat stress was breed-dependent, with abdominal and intermuscular fat deposition enhanced in one type compared to the other.

Cold stress has the opposite effect. Animals increase the oxidation of lipids for energy to maintain body temperature. Soren [51] extensively reviewed the literature associated with cold stress. In summary, the review found exposure to cold stress caused an increase in dietary intake and catabolism of fat reserves to divert energy to the generation of body heat.

The gut microbiota are also an environmental factor that regulates fat storage. Bäckhed et al. [52] utilized germ-free mice models to evaluate the impact of gut microbiome on metabolic activity. This work found that the composition of the gut microbiome impacted the energy harvested from the diet and energy storage. Maltecca et al. [53] summarized a series of studies evaluating the ability of the gut microbiome to predict growth and carcass composition. They concluded the gut microbiome could be incorporated into genomic predictions.

### 2.3. Intramuscular Fat or Marbling

Intramuscular fat or marbling is located in the perimysial space between the muscle fibers [54]. The storage of fat within the muscle increases the visible marbling, but the fat cells are thought to be in the areas at birth [55]. Marbling can be impacted by selection. However, the greatest impact on fat storage is energy intake above maintenance. When energy requirements for growth are reduced as the animal approaches mature size, the extra energy will be stored as fat both within the muscle and in subcutaneous fat. Fat distribution, subcutaneous and intramuscular (marbling), is believed to contribute to tenderness [56]. Researchers have reported increased marbling as the animal gets older [54], when fed at a similar level over time, which reflects altered maintenance requirements as the animal matures. Energy intake is still necessary to have the fat stored, including the intramuscular fat.

Furthermore, researchers using the USDA Quality grade system have reported differences in tenderness between steaks from Select and Choice carcasses, but not between the different Choice categories (Low, Average, and High Choice) [57]. As seen in carcasses from Hanwoo, Japanese Black, and Wagyu breeds, high marbling levels have been reported to be more tender [54,58]. However, other researchers have reported no difference in sensory tenderness or Warner–Bratzler shear force when comparing steaks from different quality grades [59]. Blumer [60], Jeremiah [61], and Aalhus et al. [62] found that both marbling and the subcutaneous fat cover are responsible for a small number of differences in tenderness. These studies reported that marbling might explain 2–16% of the variation in tenderness and up to 16% of the variation in juiciness [60,61]. Aalhus et al. [62] also concluded that backfat thickness had little influence on quality traits. These studies show that marbling and subcutaneous fat cover may be responsible for a small number of differences in tenderness. However, Smith and co-workers [63] concluded that using marbling and subcutaneous fat to improve tenderness was not useful. Contradictions to this have been reported when high levels of marbling in Japanese Black [54] were evaluated. Additionally, Nishimura and colleagues [54] reported a change in the connective tissue structure (electron microscopy) as the fat level in the longissimus increased. This suggests that comparisons at higher levels of marbling may be confounding the information on how marbling affects tenderness.

Marbling also affects juiciness and flavor. Small levels of marbling differences in pork result in higher sensory juiciness scores [64,65,66]. Similar results have been reported for the juiciness of the beef. McBee and Wiles [67] saw increased sensory scores for beef steaks with more marbling, with differences being more likely at lower marbling scores, suggesting a minimum level of marbling is needed for juiciness differences to be detected. Corbin and co-workers [68] compared ten different quality categories with various levels of marbling. Consumer liking for flavor and juiciness increased as the fat percentage increased. Thompson [69] found that adjusting juiciness and flavor data for similar peak shear forces resulted in a positive curvilinear relationship with an intramuscular fat percentage that plateaued at higher levels of intramuscular fat percentage. Species flavor is related to the fatty acid composition of the fat. Iida et al. [70] reported an increase in umami flavors as the fat content increased in beef from Japanese Black steers. Corbin et al. [68] also reported increased umami flavors with increased marbling and suggested that the fat level was the primary driver of beef flavor acceptability.

Despite marbling having a minor influence on tenderness, it continues to be an accepted indication of meat quality [19]. Polkinghorne and Thompson [71] reviewed meat standards and grading worldwide. They identified six different countries that utilize a marbling score as a portion of their grading system. Beef quality grades in the United States are based on physiological maturity, firmness, texture, the color of the lean, and the amount and distribution of marbling [72,73]. USDA quality grade has four main categories for young carcasses (Standard, Select, Choice, and Prime), but also divides the categories into high and low, or high, average, and low. The Canadian system includes similar characteristics to the USDA quality grades. However, they have only four quality grades for young cattle, Prime, AAA, AA, and A. Japan and Korea include many of the same characteristics in their grading system. However, there are higher marbling scores than what is seen in the US and Canadian systems [71].

Marbling is viewed as one of the most important factors influencing beef quality and palatability characteristics, especially in Hanwoo cattle, a breed that originates from Korea, and Wagyu (Japan) cattle [74,75]. Compared to European breeds, carcasses from Wagyu cattle were observed to be of a similar weight, but had significantly different carcass compositions at 24 months of age. Wagyu carcasses had a higher intramuscular fat content, 23.3% for Wagyu cattle, compared to only 0.6–4.7% for the European breeds [19]. Wagyu are also known for higher monounsaturated fatty acids (MUFA) and a higher MUFA/SFA ratio than other breeds [76], leading to the possibility of a slight health benefit for human consumption of higher-marble beef. Bessa and co-workers [77] conclude that the way to maximize CLA’s in ruminant meat was to increase intramuscular fat.

### 2.4. Development of Adipose Tissue

The primary tissues in the body are generally formed early in development from three different layers formed during gastrulation [78,79]. The three germ layers that form the other organ systems of the body are endoderm, ectoderm, and mesoderm [78,79]. Mesoderm develops into the muscle, fat, and connective tissue associated with muscle and bone. Adipose tissue develops from preadipocytes. The preadipocytes transmute to adipocytes when lipids are stored within the cell. Energy over requirements is processed into lipids stored in adipocytes. Pluripotent cells have been isolated from muscle that may play a role in the initiation of marbling [25]. De Angelis et al. [80] and Minasi et al. [81] identified myogenic cells that did not originate from the somites. This plasticity of cells could impact how muscle regenerates and influence marbling development. Harper and Pethick [55] hypothesized that several forms of stem cells could be progenitors of marbling adipocytes. This pool may replenish with cells from other parts of the body that contribute to marbling. Other mesenchymal-like cells have been isolated from muscle connective tissue [82]. The formation of the adipocytes and storage of lipids involves a highly complex, orchestrated cascade of gene expression. Lowe and co-workers [83] have constructed an integrated pathways picture that depicts this complex process.

Adipose-derived mesenchymal stem cells can differentiate into a variety of cell types; adipocytes, osteoblast, chondrocytes, and myocytes. Until recently, stem cells from adipose tissue were typically isolated as pools of mixed cell types. The ability of these isolated cells to develop into mature adipose tissue was variable [84]. More recently, surface markers have been identified on cells from this mixed population that will become functional adipocytes in adipose depots [84,85,86]. Improving adipocyte function or modifying replacement or function could benefit common metabolic diseases [83]. Spalding et al. [87] estimated that 10% of adipocytes in a human turn over per year. However, even with sustained weight loss, adipocyte numbers do not change significantly [87].

Cell to cell interactions and signal transduction are important to coordinate embryonic development. Surprisingly, there are seven major cellular pathways responsible for most animal development: Hedgehog (Hh), wingless related (Wnt), transforming growth factor-β (TGF-β), receptor tyrosine kinase (RTK), Notch, Janus kinase (JAK)/signal transducer and activator of transcription (STAT), and nuclear hormone pathways. These pathways activate specific target genes by regulating transcription factors [88].

Morrison and Farmer [89] suggested that adipocyte differentiation depends on changes in expression levels of over three hundred known proteins. Sonic Hedgehog, Wnt, C/EBP, STAT 5, Notch, PPARγ, FOXO, and SREBP are just a few transcriptional factors involved in fat accumulation (Figure 2). Other control mechanisms interact with these during development. Sonic Hedgehog (SHH) is one of three hedgehog proteins. SHH is produced as a precursor protein that promotes cell differentiation after going through proteolytic cleavage and lipid modifications [90]. It has been observed that a decrease in SHH is necessary, although not enough alone to trigger adipocyte differentiation [91]. The mechanisms controlling SHH are still poorly understood.

Ligand-receptor interactions control peroxisome proliferator-activator receptors (PPAR). There are three different types of PPARs: PPARγ, PPARα, and PPARδ; these all have other transcriptional functions [89]. PPARγ expression promotes the transcription of CCAAT/enhancer-binding proteins (C/EBP); together, these two transcriptional factors turn on lipid synthesis and other adipocyte functions. C/EBP’s are expressed in both white and brown adipose tissue and play a critical role in adipocyte differentiation [88]. C/EBP transcription factors are similar in amino acid sequences to the leucine zipper DNA-binding domain. PPARγ and C/EBP together can turn on lipid synthesis. PPARγ alone can also stimulate adipocyte differentiation; however, C/EBP cannot activate lipid synthesis without the assistance of PPARγ. The Wnt/β-catenin signaling pathway is dependent on the presence of β-catenin to proceed forward. Wnt proteins are secreted glycoproteins that are important in developing several different tissue types and cell types. Wnt plays a role in activating β-catenin and down-regulates PPARγ (Figure 3).

β-catenin regulates the expression of Pax3 in skeletal muscle and is a cofactor of forkhead transcription (FOXO) [90], binds directly to FOXO, and enhances the transcription activity of FOXO in mammalian cells [92]. FOXO proteins are a transcription factor family that binds to a specific DNA domain, the ‘Forkhead box.’ These proteins are essential because they are involved in several critical cellular processes, such as apoptosis, cell-cycle progressions, and oxidative stress resistance. FOXO target genes are also involved in glucose metabolism, cellular differentiation, muscle atrophy, and energy homeostasis [93].

Signal transducers and activators of transcription (STAT) are cytoplasmic proteins activated by gene expression in response to multiple polypeptide ligands [94]. STAT 5 up-regulates differentiation and adipogenesis [89]. Although the involvement of STAT 5 during differentiation is not well understood, it has been noted by Morrison and Farmer [89] that a lack of STAT 5 results in animals with much less white adipose tissue compared to animals with normal levels of STAT 5.

Notch signaling is another pathway involved in the regulation of adipogenesis [90]. The expression of Notch was identified in an early study as necessary for adipogenesis and was involved in the commitment of 3T3-L1 cells to go into adipogenesis [95]. However, in 2004, Nichols and colleagues found that the Notch pathway was unnecessary for adipogenesis [96]. Additionally, Ross and co-workers [97] found that Notch can regulate adipogenesis in vivo; this does not only affect the level of fat cell development, it also affects where the adipocytes form. The results from these studies are contradictory to each other. Differing results could be due to different treatments and cellular conditions [90]. This indicates that the role of Notch signaling in adipogenesis is very complex.

Sterol regulatory element-binding proteins (SREBP) are essential helix-loop-helix proteins that bind to E-box and non-E-box DNA sites [98]. An E-Box is an enhancer box; this is a DNA response component that acts as a protein-binding site. It has been reported to regulate gene expression in neurons, muscles, and other tissues [99]. There are three different types of SREBPs, SREBP-1a, SREBP-1c, and SREBP-2 [89].

### 2.5. Nutritional Restriction

Fat accretion can be strongly influenced by nutrition, whether in conditions of feed restriction or excess nutrients. During the fasting of a pregnant animal, the growth of the fetus in cattle can slow down in the second half of gestation, resulting in lighter birth weights [100]. Restricted nutrition postnatally has been shown not to negatively affect the quality of the meat in terms of shear force, compression, cooking loss, or color [100].

Many different researchers have shown a high plane of nutrition to result in more fat in most carcass depots [1,15,101,102,103]. Increased time on feed contributes to fat content, especially in beef cattle [104]. Vestergaard and co-workers [102] reported increased marbling in Friesian bull calves fed a concentrate-based diet to finish the animals compared to pasture-finished animals. Díaz and co-workers [103] found similar results in lambs fed concentrate diets, exhibiting increased fatness compared to field-finished lambs. A diet high in carbohydrates, beyond that needed for maintenance or growth, will stimulate lipogenesis in the liver and the adipose tissue, leading to high levels of triglycerides in the postprandial plasma [105]. When there are nutritional restrictions postnatally, the animal will go through a phase of compensatory gain when nutrients are returned to normal. The increased rate of gain in cattle going through compensatory gain is due to an altered maintenance requirement. Since the animal received limited nutrition, it had adjusted to survive on the available feed. Therefore, when the animal returns to a level more in line with their requirements, there is more energy than the maintenance system is accustomed to [106].

Drouillard et al. [107] found that restricted animals, in a mild and brief energy restriction, went through compensatory gain compared to animals on a consistent diet. This research found that animals with a restricted diet and animals on a normal diet finished similarly to each other, likely due to the compensatory growth of the restricted group. Jones et al. [108], Kristensen et al. [105], and Therkildsen et al. [109,110] showed that cattle and pigs being fed ad libitum feed after a period of feed restriction experienced compensatory growth as a result of increased protein synthesis and degradation. However, it is thought that protein synthesis increases at a faster rate than protein degradation. Still, with time, degradation will exceed the amount of degradation observed in animals that have not gone through nutritional restriction, leading to the increased tenderness of the product [108,109,110]. GH and IGF-1 concentrations are involved in compensatory gain as well. Plasma GH concentrations have been observed to increase in nutritionally restricted animals due to a lower nutrient influx, reducing the release of somatostatin by the hypothalamus. This minimizes the adverse effects on the synthesis and release of GH [111]. However, decreased plasma levels of hormones such as insulin lead to decreased GH-binding proteins, reducing GH attachment to receptors [112]. The increased concentration of GH and reduced amounts of insulin result in faster rates of fat mobilization. The released fatty acids provide the animal with energy [113].

Fasting can decrease lipogenesis in adipose tissue and increase lipolysis, leading to the release of triglycerides. However, triglyceride synthesis increases in the liver due to increased fatty acids from the adipose tissue. Continued high levels of triglycerides in the blood can result in a fatty liver [113].

Caroll et al. [114] evaluated carcass tissue growth in steers with a restricted energy and protein intake. Steers supplemented with grain had increased muscle and bone growth, while steers not supplemented had increased bone, but not lean growth. Neither had increased fat deposition. However, control animals that were never restricted did have more intramuscular fat than either treatment on a restricted diet. Chay-Canul and colleagues [115] evaluated the effect of a differing metabolizable energy intake (MEI) on fat depots in adult Pelibuey ewes. Animals that had consumed medium and high levels of MEI had higher levels of internal fat with respect to carcass fat than animals on a low MEI diet. More specifically, omental, pelvic, and subcutaneous fat were seen to have a dramatic increase in animals fed an energy–protein supplementation and grazing [115]. These studies indicate how differing energy consumption levels may alter fat deposition in livestock.

Hornick and colleagues [116] published a review paper on compensatory growth and the effect on carcass tissues. These researchers stated that when growth rates were reduced in response to reduced nutrient intake, there was a coordinated decrease in tissue turnover. However, tissues respond differently (viscera, adipose tissue, muscle). Fat deposition is more affected by nutrient restriction than protein deposition. Thus, the body becomes leaner [116]. Muscle growth is close to zero if fed at maintenance levels, but fat mobilization continues, and visceral weights change markedly. This can lead to altered body composition [116]. Severe feed restriction and weight losses are characterized by a sharp decrease in protein synthesis compared to degradation, indicating that the synthesis mechanisms are much more sensitive to a low (and high) feeding intensity than degradation. During nutrient restriction, fat is mobilized dependent on the severity of the nutrient restriction, whereas the protein pool is conserved as much as possible [116].

Compensatory growth is a coordinated response to realimentation. Hormonal changes during this time mediate tissue changes. Initially, there are high plasma levels of GH in response to nutrient restriction. This increase in GH could be responsible for the increased deposition of lean tissue in compensating animals [116]. A rapidly established euinsulinic state could play a vital role in initiating compensatory growth and may alleviate the resistance of the somatotropic axis to GH [116].

### 2.6. Endocrine Functions of Adipose Tissue

Adipose tissue is not just energy reserved for future use, it is also considered an endocrine organ secreting a wide range of hormones and adipokines [117]. White adipose tissue can convert androstenedione to testosterone and androgens to estrogens by the activity of 17β-hydroxysteroid oxidoreductase and aromatase [118]. However, the contribution of sex hormones from white adipose tissue to whole-body production is low. In obese patients, hormone production from white adipose tissue may influence the sex steroid profile and influence white adipose tissue developed within the body [119]. Furthermore, inflammation has been associated with increased fat depots [117]. Adipose tissue secreted factors, or adipokines, are signaling factors involved in the regulation of key homeostasis changes by autocrine, paracrine, and endocrine mechanisms. Adipokines act in appetite and satiety, regulating body fat stores and energy expenditure, glucose tolerance, insulin release and sensitivity, cell growth, inflammation, angiogenesis, and reproduction [120].

Zhang et al. [121] first cloned and identified a gene associated with adiposity from adipose tissue with autocrine and paracrine signaling highly conserved in vertebrate species. This was later termed Leptin, which is a product of the leptin gene and is produced in several tissues in addition to white adipose tissue. The primary role of leptin is modulating food intake and energy expenditure [122]. It is also thought to limit fat storage by decreasing food intake and affecting metabolic pathways that are important in maintaining the adipose tissue [113]. The levels of circulating leptin are dependent on the amount of body fat. After it is produced, leptin is secreted into the bloodstream [123]. It may then stimulate fatty acid oxidation and inhibit lipogenesis, leading to the stimulated release of glycerol from adipocytes [124]. Leptin has also been linked to inflammation in the adipose tissue associated with and demonstrated to induce insulin resistance [125,126,127]. More research is needed to better understand the roles that leptin plays in the development of the animal and the relationship between leptin, inflammation, and insulin resistance in animals.

Adiponectin is an adipokine associated with lipid trafficking and glucose homeostasis and plays a proposed role in insulin resistance and diabetes in human patients [120]. Adiponectin and the associated receptors increase AMPK and peroxisome proliferator-activated receptor α (PPARα) activity [128]. In addition, adiponectin increases insulin sensitivity by increasing the phosphorylation events in the insulin signaling cascade, and inhibits muscle and liver triglyceride deposition by increasing beta-oxidation activity [129,130,131]. Overall, adiponectin is an adipose-tissue-secreted hormone that has an inverse relationship with insulin resistance, adiposity, and inflammation markers and is an indicator of glucose tolerance. In dairy cattle, the adiponectin concentration in the dry period was negatively associated with the body condition score (BCS) and positively associated with insulin responsiveness [132].

Resistin is an adipocyte-secreted hormone that was first found to be upregulated in adipocyte differentiation, but was later found to be secreted by white adipose tissue [133]. The concentration of circulated resistin is higher in both genetic and diet-induced obese mouse models [133,134]. In both mouse and human models, resistin is associated with insulin resistance. In humans, resistin is shown to be secreted from adipose tissue depots, but is not produced in adipocytes; rather, it is produced and secreted from adipose tissue-associated monocytes and macrophages [135]. Overall, elevated resistin concentrations are associated with a reduced insulin sensitivity, hyperglycemia, and increased free fatty acid concentrations in rodents [135,136]. Limited research on resistin exists in livestock models. However, Reverchon and colleagues [137] found high plasma resistin concentrations after calving in dairy cows. This was associated with high non-esterified fatty acids (NEFA) in circulation and increased lipolytic gene expression in tissue explants.

Additional endocrine active compounds shown to be produced in adipose tissue include Apelin, Visfatin, Omentin, Vaspin, Retinol binding protein, WISP-1, Adipolin, Subfatin, and other cytokines [120]. Most of these have been identified due to their roles in the pathogenesis of human metabolic syndrome. Additional research is needed to understand the dynamics of these compounds in livestock metabolism.

## 3. Conclusions and Directions for Further Research

Adipose tissue is dynamic and is responsive to and responsible for a wide variety of hormonal and metabolic interactions with other tissues and organs, including skeletal muscle. Further research is needed to understand the complex cellular communication between adipose tissue and muscle during growth and development. In addition, the role of adipose tissue immune cells and inflammatory markers may play important roles in regulating adipose tissue deposition and in the alteration of glucose homeostasis and insulin resistance seen in fattening meat animals. An increased understanding of these complex regulatory alterations that occur when the animal body composition is changing may improve our prediction of meat quality and consistency and allow improved selection for animals that will achieve a desirable meat quality endpoint, as well as identifying animals that are better suited for specific dietary and management scenarios.

## Figures and Tables

**Figure 1 animals-12-01550-f001:**
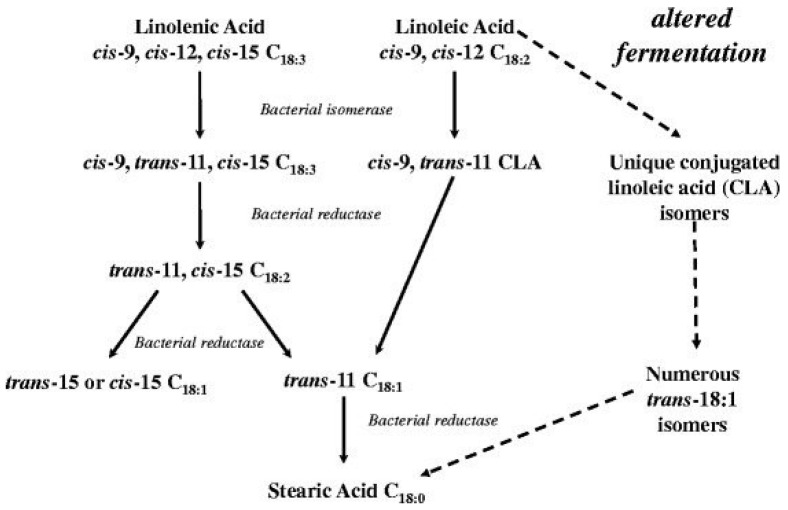
Biohydrogenation pathways of linolenic, linoleic, and acids [25].

**Figure 2 animals-12-01550-f002:**
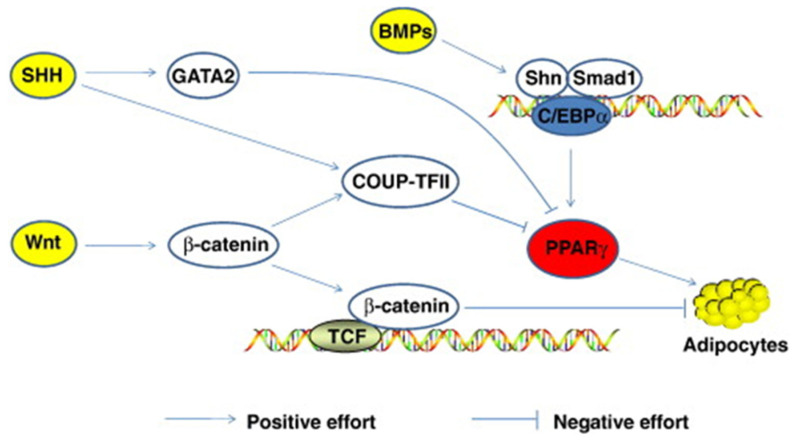
Mechanism for SHH and Wnt signaling. COUP-TFII binds to PPARγ and C/EBPα to inhibit expression [23].

**Figure 3 animals-12-01550-f003:**
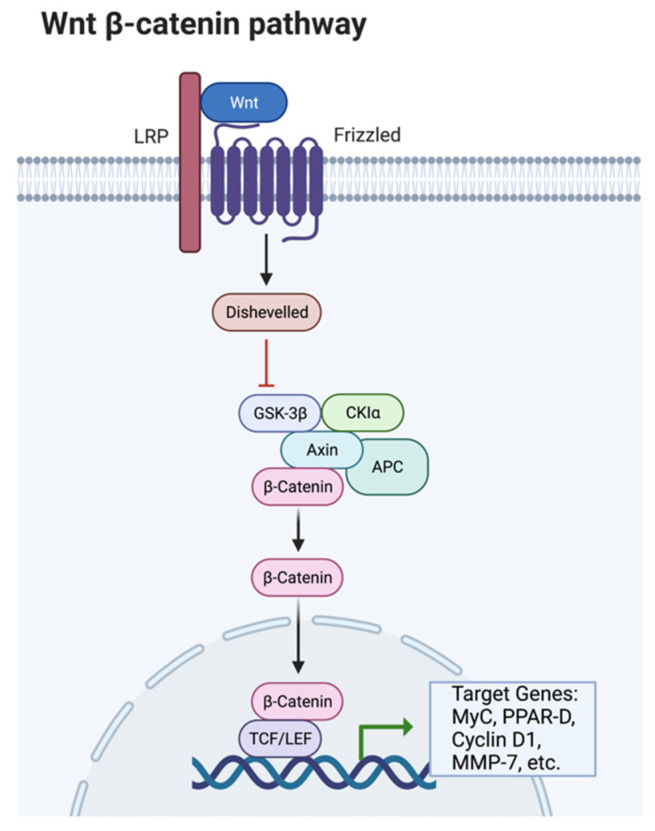
The Wnt/β-catenin pathway. Produced in biorender and adapted from cellsignal.com, accessed on 13 June 2022.

## Data Availability

Not applicable.

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
