# Peer review of "Fat Deposition and Fat Effects on Meat Quality—A Review"

_animals, 2022, doi:10.3390/ani12121550_

Round 1

Reviewer 1 Report

The manuscript entitled "Fat deposition and fat effects on meat quality – A review" summerized the relationships between animal fat deposition and meat quality, the manuscript is well organized and well written. However, some citatios were directed but not commented, so I think the authors can summerized in your one words. Besides that, the references 56 and 75 should be corrected.

Author Response

Response to reviewers is attached

Reviewer 2 Report

Dear Authors,
the review you are presenting addresses an important aspect relating to fat and its effects on meat quality. The manuscript addresses the main factors affecting the fat content and the effects on its quality, however I think there is an underlying problem with the references used. In fact, of the 124 references taken into consideration, only 71 references are after 2000 and of these only 18 from the last decade. A Review must report concise and precise updates on the latest progress made in a given research area, consequently the use of an updated references is essential for the completeness of the papers.

- The abstract must be improved by trying to objectively represent all the points reported in the paper;

- The subdivision into paragraphs should be improved;

- Lines 61-67, the paragraph relating to the influence of the species appears too reductive and not exhaustive with respect to the body of the manuscript. In my opinion the authors should improve this paragraph by implementing the discussion on the main factors related to the species that influence fat, starting with the subdivision into ruminants and non-ruminants;

- Lines 48-50, the sentence must be improved and the references updated. Recent studies have highlighted the influence of gender, age and diet on the centesimal composition, and therefore also on the fat content, of meat (for example https://doi.org/10.1080/19476337.2020.1762746);

- Lines 178-197, the paragraph should be improved by adding information relating to the influence of gender, age and stage of fattening, and therefore of the influence of the various adipose tissues, on the fatty acid composition. Finally, considering that the following paragraph (lines 201-256) exposes the influence of the diet on the fatty acid composition, the two paragraphs should be merged into a single paragraph;

- Lines 210-212 and 234-236, Alabiso et al. 2020 (https://doi.org/10.1080/19476337.2020.1842503) and other Authors have highlighted the influence of grazing on the improvement of the fatty acid profile in cattle using fresh pasture compared to barn finishing, improve sentences and update references;

- Figure 2, should be mentioned in the text;

- References, conform to journal indications and correct (references 56 and 26, eliminate repetitions).

Authors are encouraged to modify the manuscript by updating the references, with the suggestions inserted and with other more recent references, and improving the general layout.

Best regards

Author Response

Response to reviewer is attached

Reviewer 3 Report

The manuscript described the effects of fat on meat quality. Generally, it is well written and easy to follow, however, there are some points that need to be improved. The introduction should generally introduce the structure of this review, the following subsection should be numbered, otherwise, the reader will misunderstand the following parts are also the parts of the introduction. In my opinion, the "species differences in fat composition" contains similar content to "genetics or breed effect on fat", why don't you organize them together?

Line 114: What do you mean by "the greatest t impact on fat storage is energy intake", I don't understand.

Line 432: Consider revising like this: "This was later termed leptin, which is a product of the leptin gene and is produced in several tissues in addition to white adipose tissue"

References 1-22: the references are rather old, some of them are even more than 36 years ago. Please consider replacing the latest references. 

If there is a table to summarize the content of the effects factor on meat quality, it will be much better and easy to read through. 

As the manuscript described the effects of fat on meat quality, there are several attributes (e.g. water holding capacity, hardness, chewiness, cohesion, springiness, gumminess, and so on) that are related to meat quality. However, the authors only refer to tenderness, how fat affects other parameters merit to be introduced.

Author Response

Response to reviewer is attached

Round 2

Reviewer 2 Report

Dear Authors,

thank you for accepting my suggestions by improving the organization and readability of the manuscript, as well as updating the references.

Best regards

Reviewer 3 Report

The authors addressed most of my comments, it will be helpful if the authors mentioned in the latest line number to refer to their revision in the modified version, to avoid wasting reviewers' time finding differences from the original version